# Fusing Thermopile Infrared Sensor Data for Single Component Activity Recognition within a Smart Environment

**Matthew Burns, Philip Morrow *, Chris Nugent *** 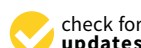 **and Sally McClean ***

School of Computing, Ulster University, Belfast BT37 0QB, Northern Ireland, UK; burns-m19@ulster.ac.uk
* Correspondence: pj.morrow@ulster.ac.uk (P.M.); cd.nugent@ulster.ac.uk (C.N.); si.mcclean@ulster.ac.uk (S.M.)

**Abstract:** To provide accurate activity recognition within a smart environment, visible spectrum cameras can be used as data capture devices in solution applications. Privacy, however, is a significant concern with regards to monitoring in a smart environment, particularly with visible spectrum cameras. Their use, therefore, may not be ideal. The need for accurate activity recognition is still required and so an unobtrusive approach is addressed in this research highlighting the use of a thermopile infrared sensor as the sole means of data collection. Image frames of the monitored scene are acquired from a thermopile infrared sensor that only highlights sources of heat, for example, a person. The recorded frames feature no discernable characteristics of people; hence privacy concerns can successfully be alleviated. To demonstrate how thermopile infrared sensors can be used for this task, an experiment was conducted to capture almost 600 thermal frames of a person performing four single component activities. The person's position within a room, along with the action being performed, is used to appropriately predict the activity. The results demonstrated that high accuracy levels, 91.47%, for activity recognition can be obtained using only thermopile infrared sensors.

**Keywords:** thermopile; infrared; sensors; activity recognition; image processing; sensor fusion; activities of daily living; computer vision; smart environments

---

## 1. Introduction

It has been predicted that the world's population will reach as high as 8.6 billion by 2030 [1]. It is also predicted that the number of people requiring 24/7 monitoring and care, whether due to a disability or an age-related issue, will also increase. Due to the detrimental psychological effects of moving into a nursing home and the fact that almost 90% of adults over 65 prefer living at home [2], it is preferable to facilitate remaining at home for as long as possible. The term, aging in place, refers to this concept and can be defined as the ability, irrespective of age or salary, to independently and safely live at home [3].

Activities of daily living (ADL) embody the day to day actions and activities that we perform independently for our own self-care. The items that fall under this category are activities such as feeding, bathing, grooming and dressing [4]. The analysis of the completion of such activities can benefit the monitoring of the health and wellbeing of residents through the detection of medical issues and lifestyle changes in addition to age-related diseases [5]. Monitoring the actions and ADL of a person in their own home provides the ability to understand their routine, which subsequently allows a better understanding of what aid is most beneficial to the person. This understanding can help facilitate the delivery of care essential for allowing a person to remain at home.

The monitoring of a home environment can be made possible through the deployment of sensors that will continuously collect relevant data and the subsequent processing of the data.

Many approaches exist which can be deployed for recognising ADL based on sensor data. In Ref. [6], an approach to ADL recognition for streaming sensor data within a smart home was proposed. Several ADL were covered in this approach, grooming, sleeping, eating, cleaning, washing and preparing meals. Sensor data were streamed and segmented into individual parts, with the intention that each segment represented the sensor events that had been triggered for a single activity. This segmentation was carried out using a sliding window where the segments were used to populate rows of training data, which the chosen machine learning model, a support vector machine (SVM), processed. The data generated from each separate sensor were separated so that each segment would, ideally, represent one activity, due to the existing knowledge of the beginning and end of sensor events triggered by the activities. This training data consisted of the activity, times for the start, duration and end of the activity and each individual sensor tag, which indicated whether the sensor had fired. The primary reason for using two continuous sliding windows was to compare the probability of correctness for each window's activity prediction. This then highlighted whether the probability trend was going up or down. To evaluate the results of the study, both five and ten-fold cross validation were implemented, producing an overall accuracy of 66%, with each activity causing a significantly visible variance amongst their individual accuracies. Activities that underachieved with regards to performance and accuracy were found to have had less training data, showing the necessity for a sufficiently large dataset.

Three popular categories of devices used to capture data are wearable devices, visible spectrum cameras and thermal infrared cameras. For example, in Ref. [7] wearable sensors were used to detect ADL, where inertial measurement units (IMUs) were used to collect and process data from actions such as sitting down, standing up, reaching high and low, turning and walking. A mock-up apartment was set up to monitor the participants' completion of a cleaning task. The task was laid out so that the participants would have to perform the previously stated actions to complete it. For example, objects were placed at various heights to force the participant to reach different heights and armchairs were placed within the environment to prompt the actions of sitting down and standing up. This allowed the system to attempt to predict the action at any given time. Each participant was required to complete the task in three, four and five minute durations. Five randomly chosen five minute trials were used for the training of the recognition algorithms, with all three and four minute trials used to test the algorithms. Participants wore a motion capture suit made up of seventeen IMUs where the acceleration, angular velocity and 3D orientation of each IMU was captured at a frequency of 60 Hz. During the task, kinematic peaks identified an activity and the activity was segmented by taking the maximum and minimum to the left and right of the peaks to estimate the activity's duration. Kinematic and angular data were extracted from the relevant body parts for each of the actions and the activities were detected and classified using the sensor signals at an accuracy of approximately 90%. The average median time difference between the manual and sensor segmentation was approximately 0.35 s. While promising accuracies were achieved in this study, wearable devices are not the preferred alternative to video sensors due to the required maintenance and the need to wear electronic equipment [8].

The use of computer vision/image processing technologies for activity recognition may provide a less invasive approach, since there is no requirement for the use of any wearable technology. The study in Ref. [9] showed that there are clear benefits to being able to incorporate image processing techniques into the task of recognising activities. Such benefits include the use of segmentation for detecting human movements or the various motion tracking algorithms facilitated by vision-based computer approaches. RGB-D cameras have also been used where depth information has been incorporated with the image data [10]. Here, the camera was positioned on the ceiling with the intention of predicting a performed action and, as a result, detecting abnormal behavior. This work considered each ADL to be predicted as a set of sub-activities or actions. A set of Hidden Markov Models (HMMs) were employed and trained using the Baum–Welch algorithm [11] to be able to accurately detect any significant changes in state. The position of a person's head and hands in 3D space were detected and recorded for the models. The three HMMs involved were configured to receive input from the head, hands and the

head and hands together, respectively. The five activities to be predicted were daily kitchen activities: *making coffee*, *taking the kettle*, *making tea* or *taking sugar*, *opening the fridge* and *other*. Here, the *other* category encompasses all other kitchen related activities. Each model individually recognised the sequence of activities and predicted the overall activity accordingly. The model that produced the highest probability for its prediction was chosen. The classification results of the experiment were produced from a test where 80 trials were used to train the model with a further 20 trials being used for testing. The model tailored to the head obtained an average f1-score of 0.80, with the model created for only the hands generating an average f1-score of 0.46. Finally, the model that made use of data of both the head and hands obtained a 0.76 average f1-score. Visible spectrum cameras, however, can give rise to a level of discomfort within the home space, due to their obtrusive nature. This can bring about a lack of natural behavior from the home's inhabitants. While they allow for the collection of useful and rich data, these security and privacy concerns have previously been highlighted by those who are subject to monitoring [3]. Such concerns can act as a roadblock for the successful production of activity recognition systems built with obtrusive elements. These concerns must be addressed.

An unobtrusive alternative to cameras that operate on the visible spectrum are devices that make use of thermal imagery or data. In Ref. [12] a thermal sensor was used to classify various postures and detect the presence of a person. A method of background subtraction was implemented where a threshold value was used to remove any pixels that were not associated with the person in the environment. The data collected when nobody was present in the environment made up a class label of its own. This data was used to calculate the threshold. The features that were extracted from the data included the difference between the threshold and the highest detected temperature, as well as the number of pixels with values larger than the threshold. The total, standard deviation and average gray levels from the pixels that made up the person were also calculated. The classification of the data was conducted by decision tree models built using Weka's J48 supervised learning algorithm. The training dataset was generated from data collected over three days and, based on 10-fold cross-validation, the model achieved 90.67% and 99.57% for pose and presence recognition, respectively. The two test datasets were generated from data collected on two separate days where the first test dataset produced 75.95% and 99.94% for pose and presence recognition, respectively. Accuracies of 60.06% for pose recognition and 91.65% for presence detection were achieved with the second test dataset. It was found that the results for the second set of test data suffered as the data were captured at a higher room temperature. It was concluded that a greater variety in the training data with regards to a larger range of ambient temperatures was required to improve the overall level of performance.

The thermopile infrared sensor (TIS) [13] can be used to detect sources of heat, for example, a person. The collected data can then be output as a grayscale image. The image produced shows only areas of heat using a range of the pixels with the highest gray levels, with the lower grey level pixels signifying cooler areas. Intricate features of heat sources cannot be distinguished due to this lack of detail and resolution in the images and, therefore, no discernable characteristics of people are captured. In the work proposed in this paper we have used two TIS devices, situated to capture two perpendicular planes. One of the devices was positioned on the ceiling of the environment and one on a tripod, surveying a side view. The captured frames of the space were analysed to attempt to predict the activities being performed by the person in the room at any given time. The analysis process involved predicting the action of the person in each frame using a collection of training data. The prediction, along with the person's proximity to known objects in the room such as the fridge or a table, was used to infer the likely activity.

This work aims to recognise single component activities including *opening and closing the fridge*, *using fridge*, *using the coffee cupboard* and *sitting at the table*. These activities were chosen as they are common sub-activities of ADL, such as making a coffee or a meal. This allowed us to investigate whether the TISs would eventually be usable for multiple component activities. This aim was fulfilled whilst sufficiently addressing any privacy concerns with regards to the capturing of images within

the home. The advantageous factors of these image processing techniques are intended to produce an accurate and unobtrusive activity recognition approach.

The remainder of this paper is structured as follows: Section 2 provides details of the platform and methodology for activity recognition, using only the TIS; Section 3 outlines the single component activity recognition experiment which was conducted; and Section 4 presents the results of the experiment; the evaluation of the results, discussion and conclusions are presented in Section 5, together with details of potential future work.

## 2. Materials and Methods

The research in this study was carried out in the smart kitchen in Ulster University [14]. This environment was equipped with numerous sensors including two $32 \times 31$ TISs which were located on the ceiling and in the corner of the room. For this work we only made use of the TISs. The two TISs were set up as sources for the SensorCentral sensor data platform [15]. The sensor data were provided by the SensorCentral sensor data platform in JSON format. An overview of the initial stages of the implemented method is depicted in Figure 1, where the sensors have captured a person bending over in front of the fridge.

The fundamental functionality of this single component activity recognition approach is to retrieve thermal frames from two sensors of the same type and extract and fuse relevant features to predict the single component activity being performed within each frame. Upon determination of the action being performed within the frame, the object nearest to the person was calculated. This process can be viewed in the pseudo code in Algorithm 1.

---

**Algorithm 1** Pseudo code for the process of calculating the nearest object

---

1:  **SET** *nearestObjectDistanceXPlane* TO 0
2:  **SET** *nearestObjectDistanceYPlane* TO 0
3:  **For Each** *frame pair*
4:      **For Each** *object*
5:          **For Each** *proximity point*
6:              **IF** *distance between BLOB's X centroid value and proximity point's X value < X plane threshold* **AND** *distance between BLOB's Y centroid value and proximity point's Y value < Y plane threshold*
7:                  **IF** *distance between BLOB's X centroid value and proximity point's X value < nearestObjectDistanceXPlane* **AND** *distance between BLOB's Y centroid value and proximity point's Y value < nearestObjectDistanceYPlane*
8:                      **SET** *nearestObjectDistanceXPlane* TO *distance between BLOB's X centroid value and proximity point's X value*
9:                          **SET** *nearestObjectDistanceYPlane* to *distance between BLOB's X centroid value and proximity point's X value*
10:                     **SET** *nearestObject* to *object*
11:                 **ENDIF**
12:             **ELSE**
13:                 **SET** *nearestObject* to *NONE*
14:             **ENDIF**
15:         **ENDFOR**
16:     **ENDFOR**
17:  **ENDFOR**

---

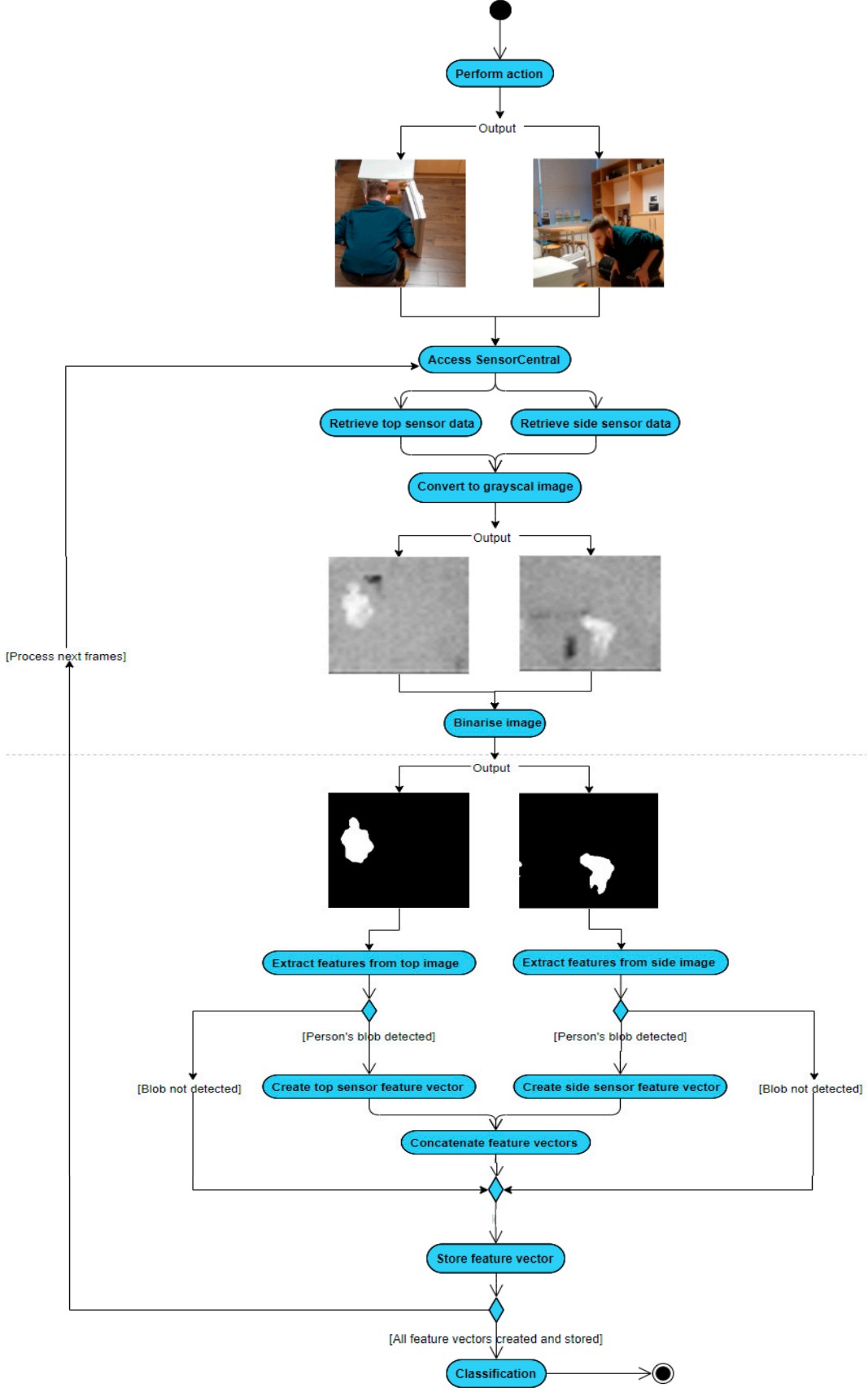

**Figure 1.** Overview of the initial stages of the method.

Once it is determined if the person is close to an object in the frame and, if so, what the object is, the object is used alongside the action to infer the activity being performed within the frame. An overview of this final aspect of the method can be viewed in Figure 2.

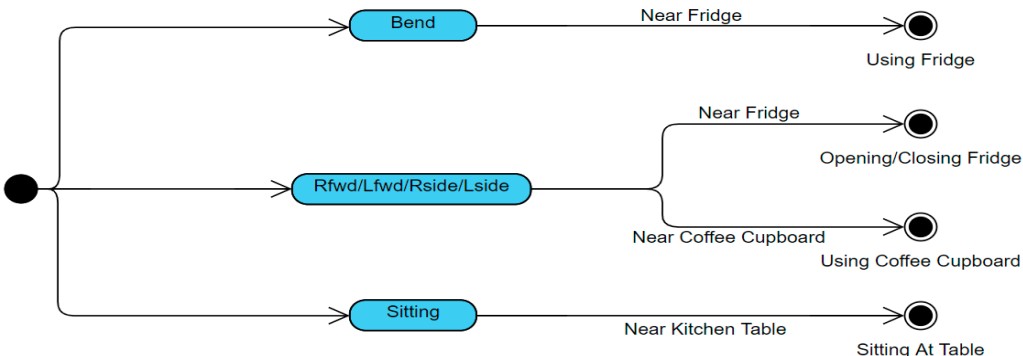

**Figure 2.** Overview of the activity inference process.

The first step in the process was to retrieve the thermal frames from SensorCentral, which acted as the middleware for the devices and the developed system. The raw data captured by the TIS were packaged in JSON format and consisted of the frame data, timestamp and sensor ID. The JSON formatted frame data from both TIS devices were retrieved and used to fill a $32 \times 32$ matrix. For convenience, the image was then resized to a $256 \times 256$ image.

The TISs were, however, $32 \times 31$ sensors and so the 32nd row is simply a black line of pixels which, when the image was resized to $256 \times 256$, made up the bottom seven rows. These rows were removed, resulting in a $256 \times 249$ image. Once the frames from both sensors were established, they were binarized using Otsu's automatic threshold method [16]. This allowed the person's shape to be analysed and features to be extracted to train the chosen machine learning model. Frames from both TISs were captured at the same time and, upon retrieval of a pair of these frames, their timestamps were compared to ensure the frames were captured at the same instant and not seconds or more apart.

The binary large object (BLOB) depicting the person was found by considering that the BLOB's area was within the pre-set parameters (chosen empirically), and that it did not have a similar centroid position to the known objects within the room i.e., the fridge, coffee cupboard and the kitchen table. Fourteen features were collected and extracted from both the shape of the person's BLOB and the pixels that made up their BLOB. The fourteen features from each frame in the pair were then combined to form a twenty-eight element feature vector. The same features were extracted from each of the sensors. The features extracted from a sensor, along with brief descriptions, are detailed in Table 1.

Since the temperature of the person may fluctuate, causing a change in pixel grey levels, features that target the person's BLOB pixel values could not be used on their own. The standard deviation and variance of the grey levels were still selected as features as they could still be somewhat useful in differentiating between the person's actions. It was, however, important to identify features that are invariant to temperature change. Performing different actions causes the shape of the person's BLOB to noticeably change and so features that describe this shape are invaluable. The eccentricity of the shape helps handle the changes in the shape's elongation and can help in detecting if the person's arms are being held out.

The convex area, equivalent diameter, solidity and extent also aid in describing the shape of the person's BLOB. This is due to the large changes that occur in the width and height of the BLOB's shape during action transitions, but also the changes in the area of the containing box or polygon when the person, for example, bends, sits or just stands with their arms down. The ratio between the major and minor axis also helps with such descriptions, where the choice to use the ratio between these values was made to create a more variable feature, making it an easier task to separate actions.

These features help differentiate between completely different actions, but it is the orientation feature that is vital to determine the difference between more similarly shaped actions such as, for example, facing a certain direction and holding the left arm out to the side and then holding the right arm to the side but facing the opposite direction. Knowing the coordinates of the bounding box encapsulating the BLOB also helps in differentiating between actions, most notably, whether it is the right arm or left arm that is being extended. The features on their own describe specific attributes of the BLOB but it is their combination that helps achieve the highest possible recognition rate.

**Table 1.** Features collected from each of the two TIS devices.

| Feature | Description |
| --- | --- |
| Eccentricity | The ratio of the distance between the foci of the shape's ellipse and its major axis length |
| Major and minor axis ratio (pixels) | Ratio between the length of the major axis of the ellipse and the length of the minor axis of the ellipse |
| Standard deviation | Standard deviation of the pixel grey levels within the detected BLOB |
| Variance | Variance of pixel grey levels within the detected BLOB |
| Bounding box corner coordinates | The coordinates of each of the four corners making up the bounding box of the BLOB, i.e., the smallest rectangle that can contain the BLOB. |
| Orientation (degrees) | Angle between the $x$-axis and the major axis of the ellipse. The value is in degrees, ranging from $-90$ degrees to 90 degrees |
| Convex area | Number of pixels in the convex hull. This is the smallest convex polygon that can contain the region |
| Equivalent diameter (Pixels) | Diameter of a circle with the same area as the region |
| Solidity | Proportion of the pixels in the convex hull that are also in the region |
| Extent | Ratio of pixels in the region to pixels in the total bounding box (smallest rectangle containing the region) |
| Moment of the shape | Returns the central sample moment of the pixel grey levels that make up the shape |

Once the features were calculated for a frame, the feature vector was stored. This was repeated until each of the frames retrieved from SensorCentral had been analysed and processed. The action being performed in each frame was manually labelled to provide ground truth data. The training dataset was made up of 3538 feature vectors, which provided sufficient examples of each action. Examples of the actions targeted for prediction are shown below, in Table 2.

Several machine learning algorithms were tried and tested to evaluate which achieved the highest accuracy of activity classification. While the support vector machine had a tendency to over fit, it was tested on the training data as it made use of what was known as a kernel trick. This technique was effective at defining clearer differences between the classes, making the process of distinguishing between them, a much simpler one. This, however, required an appropriate kernel function to be chosen. A decision tree was used as it required little intervention for any data preparation as any missing data wouldn't cause the data to split, allowing the tree to be built. The random forest machine learning algorithm was also tested as it reduces overfitting that can be caused by simple decision trees as well as bringing about less variance through its use of multiple trees.

The primary advantage to employing a random forest model for this study was its effectiveness to estimate missing data. This is a scenario that is possible, as a frame retrieved from one of the two sensors may be unusable, leaving half of the feature vector empty. This may happen due to the accidental merging of the person's BLOB with another object's BLOB or due to a sudden spike of noise injected into the frame. Using 10-fold cross validation, the random forest model achieved the best accuracy score on the training set and so was used to recognise the single-component activities performed in the experiment.

**Table 2.** Thermal frame examples from the ceiling and side sensors.

| Action | Ceiling Sensor | Side Sensor |
|---|---|---|
| *Arms down* | | |
| *Bend* | | |
| *Lfwd* (left arm forward) | | |
| *Rfwd* (right arm forward) | | |
| *Lside* (left arm extended to the side) | | |
| *Rside* (right arm extended to the side) | | |
| *Sitting* | | |

The locations of known objects within the space were also provided. These objects included the fridge, coffee cupboard and kitchen table. These objects are given what will be referred to as proximity points. The fridge and coffee cupboard had three proximity points each, located at their front left and right corners, and the middle of their south sides. The kitchen table had six proximity points, positioned at its four corners and the middle of its north and south sides. These proximity points are plotted as yellow asterisks in Figure 3 which shows the view of the ceiling TIS where the person was sitting at the kitchen table (the cyan colored rectangle). The dark blue rectangle represents the fridge, with the red rectangle representing the coffee cupboard. A compass has been annotated for reference.

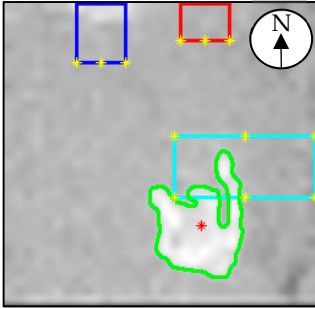

**Figure 3.** A person sitting at the kitchen table, as seen by the ceiling TIS.

The information obtained from these objects was used to determine if the person was close to any of them by measuring the distance between the person's centroid and each object's proximity points.

A diagram depicting this is shown in Figure 4, where the dashed red line signifies the shortest distance between the person's centroid and a proximity point. As this proximity point belongs to the fridge, the person is predicted to be closest to the fridge.

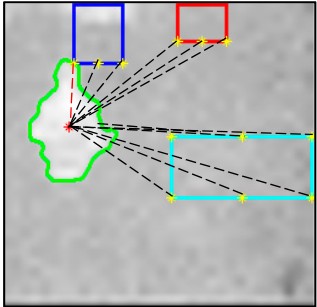

**Figure 4.** Depiction of the distance measurement between the person's centroid and each object's proximity points.

The label produced from this calculation indicates the closest object. This label was then used, along with the prediction for the performed action, to infer which class of activity was being conducted within the frame. With the action, object and activity labels populated; the original frame is annotated as shown in Figure 5.

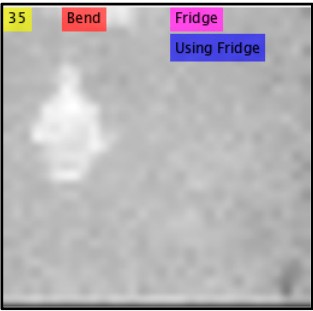

**Figure 5.** Annotated frame showing the person bending at the fridge.

The annotated image shows the frame number in yellow, the predicted action in red, the nearest object in purple and the inferred activity in dark blue. In this frame the person is predicted to be bending at the fridge and so the *using fridge* activity is inferred.

## 3. Experiment

For the experiment, each of the single component activities to be predicted were performed five times in a non-uniform order. This allowed us to adequately test the approach's capability to infer the correct activity, regardless of the order the activities were performed in. Both the TIS from the ceiling and from the side of the room were used for data capture. The thermal frames retrieved from both sensors during the performance of the activities were initially stored locally. This allowed for the opportunity to create a ground truth for each of the frames prior to processing and performance evaluation.

This ground truth was created by processing each frame one at a time, along with the pairing frame from the other TIS. The feature vectors for each frame in a pair were calculated, combined and stored. Each feature vector was then manually labelled with the action being performed, the object the person was near, if any, and the activity that was being performed, if any. This provided a ground truth state for each of the frames captured during the experiment. From each sensor, 586 frames were captured, making a total of 1172 thermal frames. There were, therefore, 586 feature vectors with a size of 28. Table 3 presents how many frames were labelled with each of the actions, objects and activities.

**Table 3.** Number of frames containing each label.

| Label | Number of Frames with Label |
|---|---|
| *Arms down* | 151 |
| *Rfwd* | 32 |
| *Lfwd* | 55 |
| *Rside* | 10 |
| *Lside* | 8 |
| *Bend* | 118 |
| *Sitting* | 212 |
| *Opening/closing fridge* | 27 |
| *Using fridge* | 118 |
| *Using coffee cupboard* | 78 |
| *Sitting at table* | 212 |
| *Near fridge* | 148 |
| *Near coffee cupboard* | 78 |
| *Near table* | 213 |

Once the ground truth was established, the accuracy of the system's action, object and activity recognition could be tested. For each frame, from both TISs, the features were extracted and combined to be passed through the trained random forest model. This produced a prediction for the action being performed.

The proximity to objects within the room was also calculated to estimate whether the person was within distance of the known position of an object that could be used. The value for the object was determined as either, *near fridge*, *near coffee cupboard*, or *near table*. The activity was inferred from both the predicted action and object values, where it could have been one of four possible activities: *opening/closing fridge*, *using fridge*, *using coffee cupboard* or *sitting at table*.

When the predictions for each of the action, object and activity values were found, they were each compared with the pre-established ground truth for that given frame to determine whether the predictions were correct. Once each frame had been analysed, total recognition accuracy for each of the previously mentioned labels was calculated.

## 4. Results

In this Section we present the accuracy of results achieved from training various machine learning models. The prediction rates for the action performed, the nearest object and the inferred activities from the conducted experiment are also broken down and evaluated.

### 4.1. Models and Overall Results

As stated previously, for each pair of frames from the two thermal sensors processed, a prediction was made for the action, the object the person was near, and the single component activity being performed. Where S1 and S2 are the frames from the ceiling and side sensor, respectively, F is the feature vector, A is the predicted action, O is the nearest object and ADL is the inferred activity. The inference is displayed in Equations (1) and (2).

$$S1 + S2 = F = A \tag{1}$$

$$A + O = ADL \tag{2}$$

For the prediction of the performed action, a machine learning algorithm was required. Of the three models tested, the random forest model, in terms of training data accuracy, achieved the best results. In Table 4, the accuracies for the action training data achieved by each model are presented. These values are based on 10-fold cross-validation.

**Table 4.** Performance accuracies based on 10-fold cross-validation.

| Model | Action Accuracy (%) |
|---|---|
| Random forest | 97.10 |
| Quadratic Support Vector Machine (SVM) | 95.20 |
| Complex decision tree | 92.90 |

The models were then used in the experiment to analyse each frame and predict the action, detect the object proximity and infer the activity. The results for the three models are shown in Table 5.

**Table 5.** Table showing results from each of the tested models.

| Model | Action (%) | Proximity (%) | Activity (%) |
|---|---|---|---|
| Random forest | 88.91 | 81.05 | 91.47 |
| Quadratic SVM | 68.40 | 81.05 | 74.20 |
| Complex decision tree | 86.68 | 81.05 | 91.29 |

The proximity accuracy does not change from model to model as it is not influenced by the approach of the chosen machine learning algorithm. The threshold to determine what is and what is not near is the only factor that plays a part in the proximity prediction. The activity prediction accuracy, therefore, varies from model to model only because the action accuracy does. It can be seen that the Quadratic SVM produces the lowest accuracy. Even though the activity accuracy achieved by the decision trees model is virtually identical to what is accomplished by the random forest, it is the improvement in the action prediction accuracy that made the random forest the best choice.

*4.2. Performed Action Results*

During the experiment there were features extracted from the shape of the person's BLOB which were used to predict the action the person was performing for that given frame. The results of these predictions for each of the seven action classes are presented in Table 6. The *Rside* action appears to be the worst performing action and has a poor recognition rate. This is inverted with regards to the *Lside* action as it was predicted correctly every time it was performed. A 100% prediction rate was almost achieved with the bend action as well as the arms down action. This differentiation between bend and arms down was made possible with the side sensor. This extra sensor data alleviated the burden on the ceiling sensor to detect differences between the two actions, resulting in the two actions rarely being confused with one another.

**Table 6.** Results for the predictions of the performed actions.

| Action | F-Score (%) | FPR (%) | FNR (%) | Precision (%) | Sensitivity (%) | Specificity (%) |
|---|---|---|---|---|---|---|
| *Arms down* | 88.00 | 7.58 | 5.30 | 82.18 | 94.70 | 92.42 |
| *Bend* | 99.15 | 0.00 | 1.70 | 100.00 | 98.30 | 100.00 |
| *Lfwd* | 87.71 | 1.89 | 9.10 | 84.75 | 90.90 | 98.11 |
| *Lside* | 64.00 | 1.72 | 0.00 | 47.06 | 100.00 | 98.28 |
| *Rfwd* | 61.76 | 2.91 | 34.37 | 58.33 | 65.63 | 97.09 |
| *Rside* | 0.00 | 0.00 | 100.00 | 0.00 | 0.00 | 100.00 |
| *Sitting* | 92.42 | 0.29 | 13.68 | 99.46 | 86.32 | 99.71 |

The low performance of *Rside* was reiterated again by the generated confusion matrix for the actions in Table 7. In this table, the row shows the true action and each column shows the action that was predicted. The rows show the actual number of instances for each action. The columns show the number of times each action was predicted, either correctly or incorrectly.

**Table 7.** Confusion matrix created from the actions predictions.

| True Class | Predicted Class | | | | | | |
|---|---|---|---|---|---|---|---|
| | *Arms Down* | *Bend* | *Lfwd* | *Lside* | *Rfwd* | *Rside* | *Sitting* |
| *Arms Down* | 143 [1] | - | - | 2 [2] | 5 [2] | - | 1 [2] |
| *Bend* | 2 [2] | 116 [1] | - | - | - | - | - |
| *Lfwd* | 2 [2] | - | 50 [1] | 3 [2] | - | - | - |
| *Lside* | - | - | - | 8 [1] | - | - | - |
| *Rfwd* | 7 [2] | - | 4 [2] | - | 21 [1] | - | - |
| *Rside* | 3 [2] | - | 2 [2] | 4 [2] | 1 [2] | 0 [2] | - |
| *Sitting* | 17 [2] | - | 3 [2] | - | 9 [2] | - | 183 [1] |

[1] Demonstrates the number of times the action was correctly predicted (true positive); [2] shows the number of times the action was predicted wrongly (false positive).

It can be hypothesised that the *Rside* performance was low due to the occlusion of the right arm from the side sensor. Throughout the experiment the right and left arms were only ever extended out to the side when the fridge or coffee cupboard was being opened. Due to the position of the side TIS, the right arm was more likely to be occluded by the person's body, leaving the classification to only the ceiling TIS. This could be addressed by capturing further frames of the *Rside* action being performed to better train the ceiling sensor to classify this action on its own. The ceiling sensor may have also struggled with the *Rside* action at the fridge as the fridge was quite low to the ground, meaning the right arm was not required to extend to the side particularly far. The inference of the activity did not suffer too much from this as almost half of the misclassified *Rside* actions were classified as the *Lside* action, which resulted in the same activity being inferred anyway.

*4.3. Proximity Detection Results*

The person's distance from each object's proximity points was calculated to determine the object the person was closest to if they were within the specified threshold. The results for each object are shown in Table 8.

**Table 8.** Results for the calculations of the proximity detection for any given frame.

| Object | F-Score (%) | FPR (%) | FNR (%) | Precision (%) | Sensitivity (%) | Specificity (%) |
|---|---|---|---|---|---|---|
| *Fridge* | 87.57 | 11.38 | 0.00 | 77.89 | 100.00 | 88.62 |
| *Coffee cupboard* | 85.71 | 5.21 | 3.85 | 77.30 | 96.15 | 94.79 |
| *Kitchen table* | 90.06 | 15.21 | 0.00 | 81.90 | 100.00 | 84.79 |
| *None* | 41.94 | 0.00 | 73.47 | 100.00 | 26.53 | 100.00 |

The confusion matrix for the proximity detections that were produced from the experiment is displayed in Table 9 and shows how the *None* label was the main reason for the lowered accuracy value. The person was frequently detected as being near the objects when, actually, they were not near any of them. This, however, did not affect the accuracy of the activity inference as the proximity detection for the three objects was almost 100% accurate any time the person was actually near one of them.

**Table 9.** Confusion matrix created from the proximity detections.

| True Class | Predicted Class | | | |
|---|---|---|---|---|
| | *Fridge* | *Coffee Cupboard* | *Table* | *None* |
| *Fridge* | 148 [1] | - | - | - |
| *Coffee cupboard* | - | 75 [1] | 3 [2] | - |
| *Kitchen table* | - | - | 213 [1] | - |
| *None* | 42 [2] | 22 [2] | 44 [2] | 39 [1] |

[1] Demonstrates the number of times the object was correctly predicted (true positive); [2] shows the number times the object was predicted wrongly (false positive).

### 4.4. Activity Inference Results

From both the performed action and the nearest object to the person, the activity, if any, was inferred. The results for the prediction of the performed activity within each frame are presented in Table 10.

**Table 10.** Results for the predictions of the inferred activities for all frames captured during the experiment.

| Activity | F-Score (%) | FPR (%) | FNR (%) | Precision (%) | Sensitivity (%) | Specificity (%) |
|---|---|---|---|---|---|---|
| *Opening/closing fridge* | 80.00 | 0.58 | 25.93 | 86.96 | 74.07 | 99.42 |
| *Using fridge* | 99.15 | 0.00 | 1.690 | 100.00 | 98.31 | 100.00 |
| *Using coffee cupboard* | 94.59 | 0.00 | 0.260 | 100.00 | 99.74 | 100.00 |
| *Sitting at table* | 92.42 | 0.28 | 13.68 | 99.46 | 86.32 | 99.72 |
| *None* | 85.47 | 10.57 | 2.65 | 76.17 | 97.35 | 89.43 |

As stated, it was the results from the action classification and proximity detection from which the activities were classified. The slightly lower proximity detection accuracy did not have any significantly detrimental effect on the activity accuracy. This was most likely because the misclassifications of the nearest object were caused by the person walking past an object, as opposed to using it, and being predicted as near another. The low detection rate for the *Rside* action also did not show any significant negative effects on the activity accuracy. The confusion matrix for the activity predictions is presented in Table 11.

**Table 11.** The confusion matrix created from the activity predictions.

| True Class | Predicted Class | | | | |
|---|---|---|---|---|---|
| | *Opening/Closing Fridge* | *Using Fridge* | *Using Coffee Cupboard* | *Sitting at Table* | *None* |
| *Opening/closing fridge* | 20 [1] | - | - | - | 7 [2] |
| *Using fridge* | - | 116 [1] | - | - | 2 [2] |
| *Using coffee cupboard* | - | - | 70 [1] | - | 8 [2] |
| *Sitting at table* | - | - | - | 183 [1] | 29 [2] |
| *None* | 3 [2] | - | - | 1 [2] | 147 [1] |

[1] Demonstrates the number of times the activity was correctly predicted (true positive); [2] shows the number of times the activity was predicted wrongly (false positive).

## 5. Discussion and Conclusions

This aim of this paper was to propose an unobtrusive and accurate approach to single component activity recognition. The study involved evaluating the use of two TISs for activity recognition where it was found that the introduction of the second sensor benefited the accuracy of using only TIS device types for activity recognition. We captured data for seven different actions to train various machine learning models, where the random forest achieved the highest accuracy. The positions of three objects within the kitchen were noted and the action and object combinations were determined to allow for

the inference of single component activities. The trained model was tested and evaluated to determine its ability to predict the actions and, as a result, the inferred activity.

The conducted experiment allowed thermal frames to be captured to evaluate the trained random forest model. A prediction for the performed action and the closest object were used in conjunction with one another to infer if an activity was being performed in the frame. This was completed for each of the frames, where the predictions were compared with the ground truth to determine the recognition accuracy for each of the three labels. These experimental results were very good, with accuracies of 88.91%, 81.05% and 91.47% achieved for the action, proximity detection and inferred activity, respectively. With the incorporation of the side sensor, actions such as arms down and bend were easily distinguishable. The second sensor also helped avoid issues caused by image noise, making the approach more robust. When too much noise caused difficulties in detecting the person's shape, making the frame unusable for extracting features, the frame could be disposed of without concern as the second sensor's frame could still be used on its own for feature extraction.

The *Rside* action prediction underperformed with each of its ten instances being misclassified as another action. The implication of this low accuracy is, however, alleviated by the fact that almost half of the misclassifications were for *Lside*, resulting in a correctly inferred activity. This low accuracy is also in the minority as the other targeted actions were predicted with high accuracy, as shown by the 100% and 99.46% precision values for bend and sitting, respectively.

The results for the proximity detection was adequate, however, limited. The thresholds chosen for the distances in the X and Y planes proved to be appropriate for attaining the best proximity accuracy. This showed that there will be a need for refinement and further innovation in the proximity area of the work to subsequently improve upon the activity inference accuracy, potentially through the implementation of ultra-wideband (UWB) for 3D positioning of the kitchen objects. The activity inference yielded a high recognition accuracy supporting the case for the TIS device as an efficient and more than effective means for single component activity recognition within a smart environment.

This approach has, therefore, demonstrated that the advantages of image processing techniques with visible spectrum images for smart home moderation can be retained without breaching privacy and using only the TIS device. This can be facilitated through the unobtrusive collection of data, as no discernible characteristics of people are targeted, and through the automated nature, as no wearable devices are required to monitor inhabitants. There is, however, potential for even further improvement and expansion of this method.

The need for future work to enhance the proposed system has been considered. While a more extensive set of training data could improve the accuracy of the *Rside* action, the issue may be one of occlusion. The prediction rate could then be improved by implementing an eighth action class, Occluded. This label would belong to frames where the ceiling sensor's feature data would describe one action e.g., *Rside*, while the side sensor data would describe another e.g., *arms down*. In such scenarios, the frame and the feature data extracted from it would be disregarded for the inference of the performed activity.

The dataset used was not balanced for some class labels, for both training and testing, and, although relatively high accuracies were achieved, this imbalance will need to be addressed in future work. The imbalance was likely caused by the manner in which each action was captured. As a person is likely to perform each action randomly and for varying durations in a real-life scenario, the training data for a particular action were captured by performing that action in a similar vein. For example, if a five minute time limit was used to capture some data for the *Lside* action, the person would perform this action in different parts of the room for different durations. The intention was that the training data would be made up of actions being performed in more realistic scenarios. This resulted in the data including frames of the person doing movements other than the targeted action such as walking and performing the arms down action.

For the classes in the testing dataset, the experiment involved completing the activities five times each with no given time limit for the activity performance. This meant that the time spent

on each activity was not necessarily equal, resulting in some actions being performed more than others. This inequality was also likely caused by some actions not being necessary for some activities, for example, *sitting* was not required for *using fridge*. A more balanced set of training data, however, may produce an even more accurate recognition rate. The approach to capturing training data in future work will therefore be stricter and more aimed toward a balanced class size rather than the recreation of a real-life scenario.

The system described in this study will be expanded upon in the future to not only recognise sub activities but also the ADL they make up. This will require an understanding of which sub activities make up each targeted ADL and which actions signal their beginning and end. It will be vital to facilitate the tracking of the performed sub-activities over time to analyse the several activities that encompass the ADL performance, as opposed to the single frame analysis that is demonstrated here. It will also be important to incorporate, for example, a Bayes statistical model to apply probabilities to each of the activities potentially being performed. This will allow for evidence to be built over time to better determine the likelihood of an activity being performed. Different combinations of the list of extracted features may also be examined with the intention of efficiently improving the prediction rate of activities within a smart environment. Further sensor fusion approaches will be investigated, potentially involving other sensor types.

Maintaining privacy for inhabitants of smart environments remains an important factor in ADL analysis. Due to this, regardless of the future work that is conducted to improve upon the findings of this study, the preservation of the system's unobtrusive nature will remain a priority.

**Author Contributions:** Conceptualisation, M.B., P.M., C.N., and S.M.; data curation, M.B.; methodology, M.B., P.M., C.N., and S.M.; software, M.B.; supervision, P.M., C.N., and S.M.; writing—original draft, M.B.; writing—review and editing, M.B., P.M., C.N. and S.M.

**Funding:** This research was supported through a Northern Ireland Department for the Economy (DfE) PhD scholarship.

**Conflicts of Interest:** The authors declare no conflict of interest. The funders had no role in the design of the study; in the collection, analyses, or interpretation of data; in the writing of the manuscript, and in the decision to publish the results.

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
