# Peer review of "Fusing Thermopile Infrared Sensor Data for Single Component Activity Recognition within a Smart Environment"

_jsan, doi:10.3390/jsan8010010_

Reviewer 1 Report

This paper is devoted to activity recognition within a smart environment. The visible spectrum cameras used as data capture devices. The paper claims that high accuracy levels of 91.47% for activity recognition can be obtained when using Thermopile Infrared Sensors only.

My only question refers to this statement: “Fourteen features are collected and extracted from both the shape of the person’s BLOB and the 191 pixels that make up their BLOB”.  Why were these features chosen? Usually, this is one of the most important (and most interesting) moments in machine learning. It would be interesting to see the explanation.

Author Response

Please find the response in the uploaded file.

Reviewer 2 Report

Overall a well-written, easy to understand article.  The results and conclusions fit together and were supported by the data.

I did not see anything particularly novel to this approach. Perhaps the authors could clearly highlight their contributions

The class labels exhibit a large amount of imbalance with some labels having orders of magnitude more examples than other classes.  A more balance dataset might lead to better results. This should be discussed more directly in the paper.

Author Response

(The authors gave the same response as above.)
